# Paired electrocatalysis unlocks cross-dehydrogenative coupling of C(sp³)-H bonds using a pentacoordinated cobalt-salen catalyst

Ke Liu[1], Mengna Lei[1], Xin Li[1], Xuemei Zhang[1], Ying Zhang[1], Weigang Fan[1], Man-Bo Li [1] ✉ & Sheng Zhang [1] ✉

Cross-dehydrogenative coupling of C(sp³)-H bonds is an ideal approach for C(sp³)-C(sp³) bond construction. However, conventional approaches mainly rely on a single activation mode by either stoichiometric oxidants or electrochemical oxidation, which would lead to inferior selectivity in the reaction between similar C(sp³)-H bonds. Herein we describe our development of a paired electrocatalysis strategy to access an unconventional selectivity in the cross-dehydrogenative coupling of alcoholic α C(sp³)-H with allylic (or benzylic) C-H bonds, which combines hydrogen evolution reaction catalysis with hydride transfer catalysis. To maximize the synergistic effect of the catalyst combinations, a HER catalyst pentacoordinated Co-salen is disclosed. The catalyst displays a large redox-potential gap (1.98 V) and suitable redox potential. With the optimized catalyst combination, an electrochemical cross-dehydrogenative coupling protocol features unconventional chemoselectivity (C-C vs. C-O coupling), excellent functional group tolerance (84 examples), valuable byproduct (hydrogen), and high regio- and site-selectivity. A plausible reaction mechanism is also proposed to rationalize the experimental observations.

The development of approaches forming C(sp³)−C(sp³) bonds is a central topic in organic synthesis[1–5] A strategy based on cross-dehydrogenative coupling (CDC) was initially unveiled by Li[6–8], and it has been demonstrated as a robust and general tool for the construction of C(sp³)−C(sp³) bonds with hydrogen as the only byproduct. Conventional CDC approaches commonly involve stoichiometric oxidants and high temperature (Fig. 1a), which could result in some selectivity issues, including overoxidation, side reactions and undesired chemoselectivity. With the significant progress of synthetic electrochemistry[9–19], it offers alternative choices[20–23] for the redox chemistry by directly manipulating electron transfer between substrates and electrodes. Consequently, an impressive breakthrough has

been achieved in the electrochemical CDC[24,25]. However, most of the electrochemical CDC approaches only involve anodic oxidation, and nucleophiles bearing reactive C(sp³)-H are commonly required (Fig. 1b). With a long-term interest[26–29] in paired electrolysis[30–32], we envisaged that merging anodic oxidation and cathodic reduction might enable a dual activation mode for the CDC reaction of unactivated C(sp³)-H bonds (Fig. 1c), which would otherwise be inaccessible either for the conventional CDC approach or electrochemical oxidative CDC.

The cross-dehydrogenative coupling of alcohols with allylic and benzylic C-H bonds could serve as a general and direct approach to upgrade simple alcohols to value-added alcohols. Nevertheless,

[1]Institutes of Physical Science and Information Technology, Key Laboratory of Structure and Functional Regulation of Hybrid Materials of Ministry of Education, Anhui University, Hefei 230601 Anhui, China. ✉e-mail: mbli@ahu.edu.cn; shengzhang@ahu.edu.cn

**Fig. 1 | Approaches for the cross-dehydrogenative coupling (CDC).**
**a** Conventional approaches for CDC. **b** Electrochemical oxidative approaches for CDC. **c** Electrochemical CDC enabled by paired electrolysis.

previous reports for the transformation only afford C-O coupling products either with conventional[33,34] or electrooxidative[35] protocol due to the mismatch reactivity of the substrates (Fig. 2a); O-H is more acidic than C-H bond and it can be easily converted to an oxygen anion nucleophile. To reverse the chemoselectivity in the reaction, we herein reported a paired electrocatalysis[36–39] strategy (Fig. 2b), which consists of a hydride transfer (H⁻T) catalyst[40] and a hydrogen evolution reaction (HER) catalyst[41–50] enabling a dual activation mode for both substrates. In the paired electrocatalytic system, the H⁻T catalyst could selectively remove hydride from alcohols to form aldehydes via carbocations, while the HER catalyst shifts the dissociation equilibrium of weak acidic C-H bonds to deliver hydrogen and carbanions, thus affording alkylated alcohols.

To establish an efficient paired electrocatalytic system, good compatibility of anodic and cathodic catalysts is the first concern. Active anodic and cathodic catalysts have opposite redox natures, and their mutual interference would result in deteriorated catalytic efficiency. As shown in the Fig. 2c, the undesired reaction between the active species **AM⁺•** and **CM⁻•**, which respectively arise from the anodic mediator (**AM**) and cathodic mediator (**CM**), would directly reduce the overall catalytic efficiency. In this work, a pentacoordinated Co(II)-salen catalyst was demonstrated as a stable HER catalyst with a large redox potential gap and suitable oxidation potential. By cooperating with a common hydride transfer (H⁻T) catalyst 2,2,6,6-tetra-methylpiperidoxyl (TEMPO), the pentacoordinated Co(II)-salen catalyst unlocks an unconventionally chemoselective CDC reaction of

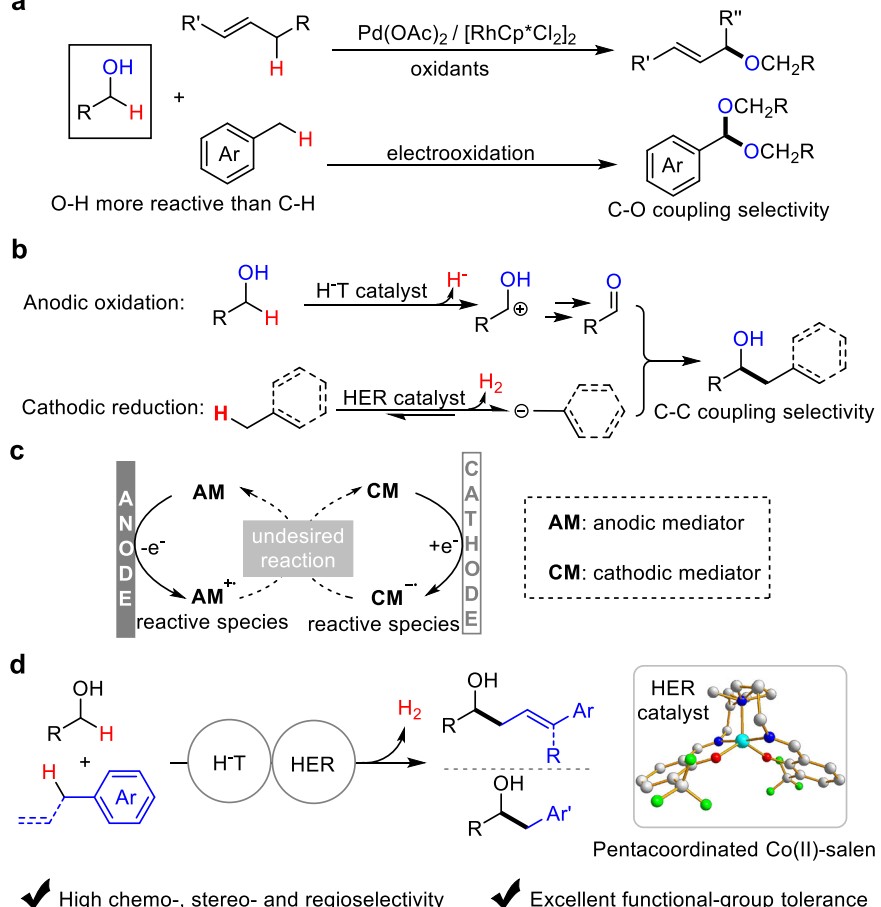

**Fig. 2 | Strategies toward the cross-coupling of alcohols with unactivated C(sp³)-H bonds. a** Previous reports for cross-coupling alcohols with C(sp³)-H bonds. **b** Using paired electrocatalysis strategy to reverse the reaction chemoselectivity. **c** Challenges in the paired electrocatalysis. **d** This work.

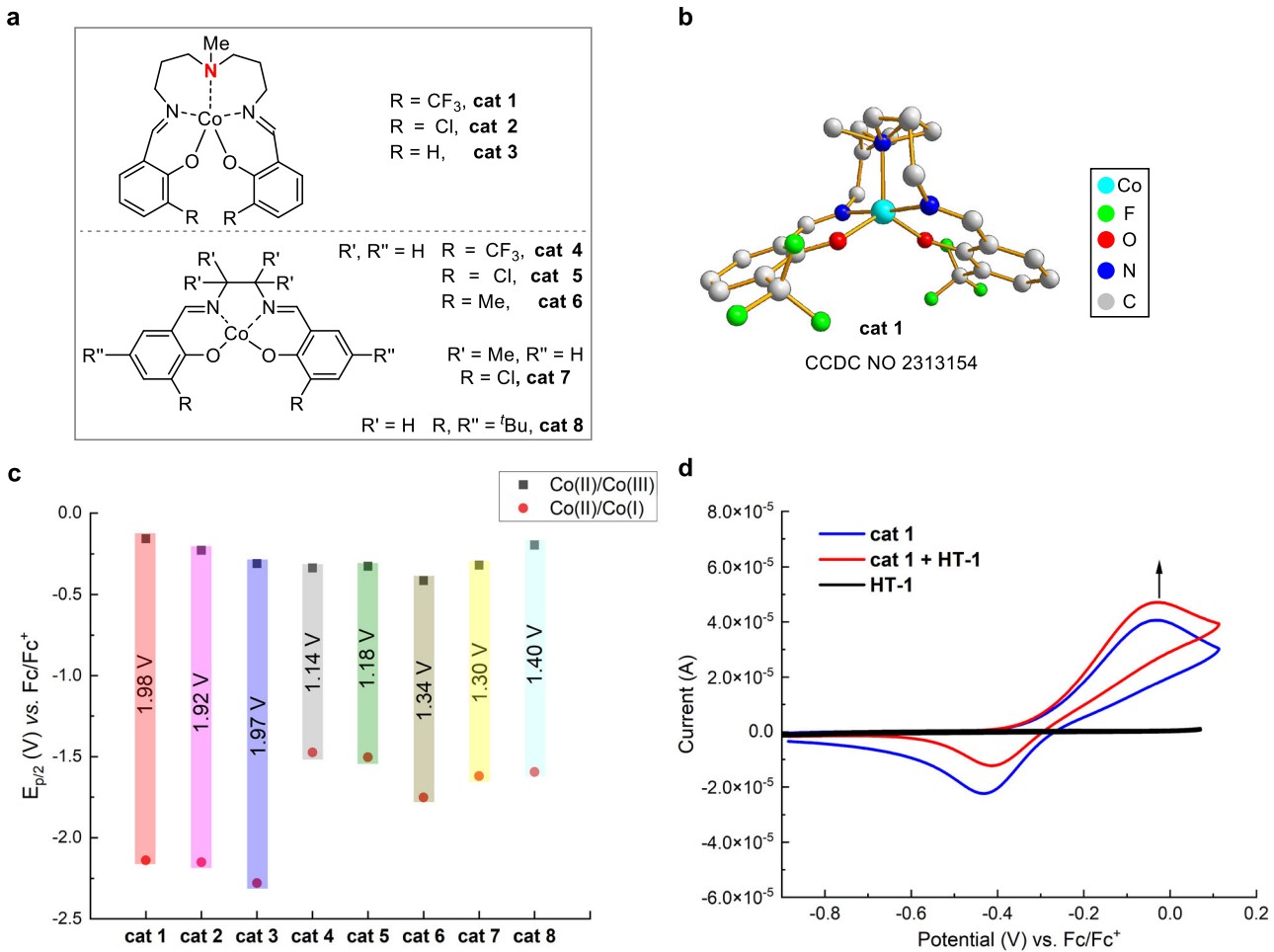

**Fig. 3 | The evaluation of HER catalysts. a** The structures of cobalt-salen catalysts. **b** Single crystal structure of **cat 1**. **c** Redox property of **cat 1**–**cat 8**. **d** Catalytic role of **cat 1** in the electrooxidation of TEMPO (**HT-1**).

alcohols with allylic and benzylic C-H bonds to open up a general platform upgrading alcohols (Fig. 2d). This paired electrocatalytic protocol features high chemo-, stereo- and regioselectivity, excellent functional-group tolerance (>80 examples) and mild conditions.

## Results

### The evaluation of HER catalysts

To better understand the relationship of structure and redox properties, we initially synthesized a series of Co[II]-salen[51,52] HER catalysts **cat 1-cat 8** (Fig. 3a). The structure of pentacoordinated catalyst was unambiguously established by single crystal diffraction analysis of **cat 1** (Fig. 3b) (Deposition number 2313154 (for **cat 1**) contains the supplementary crystallographic data for this paper. These data are provided free of charge by Cambridge Crystallographic Data Centre.), and it shows that cobalt center is shielded by a roof-like nitrogen coordination site, which would significantly enhance its stability. The redox property of the catalysts was next explored to support the above speculation. We recorded and identified all of the redox peaks of catalysts arising from the process of Co[III]/Co[II][53,54] and Co[II]/Co[I] (Fig. 3c). As expected, pentacoordinated catalysts (**cat 1-cat 3**, 1.98–1.92 V) have uniformly larger redox potential gap than that of the conventional catalysts **cat 4-cat 8** (1.14–1.40 V). This result verifies that pentacoordinated Co-salen catalysts are more stable under oxidation and reduction compared with their tetracoordinated counterparts. Noteworthy, $CF_3$ group was found to significantly improve the antioxidation stability of **cat 1** with the most positive peak (Co[III]/Co[II]) at −0.157 V (vs. Fc/Fc$^+$). This redox property endows **cat 1** with good compatibility

with the aforementioned cocatalyst TEMPO (**HT-1**). Moreover, we also compared the anodic oxidation of **cat 1** (−0.157 V) with that of **HT-1** (0.195 V); the narrow potential difference[55] made it possible to use **cat 1** to facilitate the oxidation **HT-1**. Further cyclic voltammetry experiment mixing **cat 1** with **HT-1** confirms the above conclusion by detecting an obvious increase of the anodic peak of **cat 1** (Fig. 3d). Nevertheless, other cobalt catalysts failed to produce the same results when mixing with **HT-1**. Taken together, **cat 1** showed good compatibility with the H-T catalyst **HT-1**. The synergistic effect of the catalyst combination was demonstrated by the acceleration effect of **cat 1** on the oxidation of **HT-1**[56].

### Reaction optimization

Encouraged by the redox property study, we next investigated the cross-coupling reaction between benzyl alcohol (**1a**) and allylbenzene (**2a**) with various catalyst combinations. We screened catalysts using $Cs_2CO_3$ as base additive, $^nBu_4NClO_4$ as electrolyte, N,N-dimethylformamide (DMF) as solvent, graphite felt and copper plate as anode and cathode, respectively. As shown in Table 1, a suitable catalyst combination is crucial for the reaction efficiency in terms of yields (Entries 1–11). The optimal catalyst combination was identified with **cat 1** as HER catalyst and **HT-1** as H-T catalyst, and the desired C-C coupling product **3a** was obtained in 85% yield with exclusive C-C coupling and C=C position isomerizing selectivity. By comparing the yields arising from other catalyst combinations (Supplementary Table 2), we concluded that the HER catalyst with a large potential gap and the H-T catalyst with a low oxidation peak would benefit the reaction

**Table 1 | Optimization of cross-coupling reaction between benzyl alcohol 1a and allylbenzene 2a**

| Entry | Catalyst combinations (HER catalyst + H·T catalyst) | Yield (%)[a] |
|---|---|---|
| 1 | **cat 1 + HT-1** | 85 |
| 2 | **cat 2 + HT-1** | 65 |
| 3 | **cat 3 + HT-1** | 61 |
| 4 | **cat 4 + HT-1** | 56 |
| 5 | **cat 5 + HT-1** | 57 |
| 6 | **cat 6 + HT-1** | 63 |
| 7 | **cat 7 + HT-1** | 59 |
| 8 | **cat 8 + HT-1** | 71 |
| 9 | **cat 1 + HT-2** | 38 |
| 10 | **cat 1 + HT-3** | 39 |
| 11 | **cat 1 + HT-4** | 55 |
| 12 | Only **cat 1** | 25 |
| 13 | Only **HT-1** | 33 |
| 14 | None | trace |
| 15[b] | Conventional CDC conditions | trace |

Reaction conditions: **1a** (0.5 mmol), **2a** (1.5 mmol), HER catalysts (5 mol%), H·T catalysts (20 mol%), $Cs_2CO_3$ (20 mol%), $^nBu_4NClO_4$ (1.0 mmol), DMF (10 mL), graphite felt (GF) anode, copper plate cathode, CCE = 20 mA, 4 h (5.97 F/mol), 0 °C.
*DMF N,N*-dimethylformamide, *CCE* constant current electrolysis.
[a]Isolated yield.
[b]See Supplementary Table 3 for details.

performance. The necessity of both catalysts was also supported by the control experiments removing either of the catalysts (Entries 12–14). We also highlighted the superiority of this electrochemical protocol when compared to conventional CDC conditions (Entry 15, see Supplementary Table 3 for details).

**Exploration of scope**

With the optimal reaction conditions in hand, a wide range of substrates bearing weakly acidic $C(sp^3)$-H were investigated to couple with benzyl alcohol **1a** (Fig. 4). Initially, we examined a series of toluene derivatives, and the desired C-C coupling products (**3b–3i**) were obtained with moderate yield. Notably, bioactive amide and sulfonamide substrates (**3e, 3h–3i**) bearing reactive α C-H were well-tolerated. Moreover, 4-methylbenzophenone was also amenable to give the corresponding product **3f**, which conventionally preferred to proceed a pinacol coupling of 4-methylbenzophenone. To our delight, a mixture of β-methylstyrene isomers can be directly used as substrate to afford corresponding products with high *E/Z* selectivity (**3j, 3k**). Besides the benzylic and allylic $C(sp^3)$-H, (methylsulfinyl)benzene proved to be suitable to give the desired product **3l**. Subsequently, various allylbenzene derivatives were tested in the electrochemical protocol. Substrates with multiple double bonds (**3m**) or different substitution patterns (**3n–3q**) are all well-tolerated to deliver the terminal C-H coupling products with exclusive position-isomerizing selectivity and *E/Z* selectivity. The site-selectivity of the electrochemical approach was also demonstrated by the case of **3r**, and more

acidic C-H is favored in the reaction. This cross-coupling reaction also enabled a late-stage functionalization for the natural product Magnolol and Eugenol-derived substrates (**3s–3t**). Specifically, both allyl groups underwent isomerization to give a mono-coupling product **3s** (as shown in the Fig. 4). Additionally, we investigated the electronic effect (**3u–3ad**) and position of substituents (**3ae–3ag**), fused ring (**3ah–3ai**) and heterocycle (**3aj**). Uniformly good yields were observed for the substrates, although the thienyl group led to diminished *E/Z* selectivity. It is noteworthy that radical-sensitive groups such as cyclopropyl (**3p, 3x**) and ortho-vinyl (**3ag**) were untouched during the electrochemical transformation, ruling out the possible radical pathway. Finally, toluene, 4-methyl anisole, hexene, and 4-phenyl-1-butene proved to be failed substrates in the reaction due to their less acidic C-H bonds.

To further demonstrate the generality of the electrochemical cross-coupling reaction, we next tested a broad range of alcohols (Fig. 5). In general, variations on the electronic property (**3ak–3bg**) and substitution pattern (**3bh–3bq**) of benzyl alcohols are well tolerated. Electron-deficient substrates (**3as–3ay**) with higher oxidation potential proved to be less efficient compared with the electron-rich alcohols (**3ak–3ar**). The functional-group tolerance of this approach was highlighted by the cases of **3az–3be**, which commonly cannot survive under conventional CDC conditions. Furthermore, site-selectivity favoring less-hindered benzylic C-H was observed in the substrates containing two hydroxyl groups (**3bf–3bg**). Other aromatic (**3br–3bw**), aliphatic (**3bx–3cb**), secondary (**3cc–3cd**) and bioactive

**Fig. 4 | Scope of the substrates bearing weakly acidic C(sp³)-H.** Reaction conditions: **1a** (0.5 mmol), **2** (1.5 mmol), **cat 1** (5 mol%), **HT-1** (20 mol%), Cs₂CO₃ (20 mol %), ⁿBu₄NClO₄ (1.0 mmol), DMF (10 mL), graphite felt anode, copper plate cathode, CCE = 20 mA, 4 h (5.97 F/mol), 0 °C, *E/Z* > 15/1; DMF *N,N*-dimethylformamide, CCE constant current electrolysis. The percentage listed under each of the substrate refers to the isolated yield obtained in the reaction. ᵃ*E/Z* ~ 1.4/1.

molecule (amylcinnamyl alcohol, Adapalene) derived alcohols (**3ce**–**3cf**) were also found to be suitable substrates to afford corresponding products with synthetically useful yields. Alcohols bearing cyclopropyl group gave the desired products **3bx**–**3by**. The possible radical-initiated ring expansion byproducts were not detected in the reaction mixture, further excluding the radical pathway in the transformation.

**Applications in synthesis**
Having examined the reaction generality, we turned our attention to probing the utility of our electrochemical protocol with gram-scale reaction and derivatization of products (Fig. 6). Gratifyingly, scaling up the model reaction to gram-scale afforded product **3a** in a satisfactory yield (70%) even with lower catalyst loading (Fig. 6a). The byproduct, that is, the gaseous hydrogen was successfully collected using a balloon and verified with GC. This result suggests that protocol not only provides a route for organic transformation but also enables an avenue for hydrogen. Using a simple solar cell as an electricity supply largely maintained the reaction efficiency (Fig. 6b). Under acidic conditions, the dehydration of homoallylic alcohol **3m** gives a conjugated light-emitting molecule **4** (Fig. 6c), which displayed prominent luminescent properties both in solution and solid state with a maximum emission peak at 493 nm. Additionally, bioactive tetrahydrofuran product **5** was

also accessed in high yield via *N*-iodosuccinimide (NIS)-initiated iodocyclization.

**Mechanism investigation**
To gain insight into the reaction mechanism, we conducted a series of cyclic voltammograms and control experiments (Fig. 7). First, the electrochemical properties of **1a** and **2a** were investigated (Fig. 7a, b). As shown in Fig. 7a, both **1a** and **2a** seem to be redox inert within the window of −2.5–2.5 V (*vs.* Fc/Fc⁺). After enlarging the range from −0.5 to −2.5 V, a couple of small waves (10⁻⁶ ~ 10⁻⁵ A) were detected at −2.04 V, which were attributed to the hydrogen atom absorption and desorption process in HER (see Supplementary Fig. 26 for GC detection of H₂). Moreover, their intensity increases with the concentration of **2a**. These observations illustrate that both substrates are relatively unactive in the absence of catalysts. Second, the catalytic role of **cat 1** and **HT-1** was explored (Fig. 7c, d). Treatment of **cat 1** with excess **2a** led to significant increases in both cathodic peaks of **2a** and Coᴵᴵ/Coᴵ; this observation verifies the catalytic role of **cat 1** in the HER of **2a**. On the other hand, mixing **HT-1** with benzyl alcohol **1a** also resulted in an obvious catalytic current, suggesting the catalysis of **HT-1** in the oxidation of **1a**. The titration experiments (Supplementary Figs. 24 and 25) were also conducted to further verify the catalytic

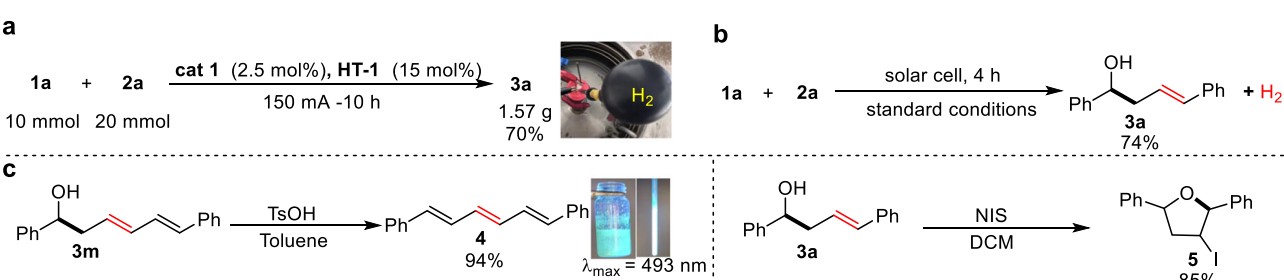

**Fig. 5 | Scope of alcohols.** Reaction conditions: **1** (0.5 mmol), **2a** (1.5 mmol), **cat 1** (5 mol%), **HT-1** (20 mol%), Cs$_2$CO$_3$ (20 mol%), $^n$Bu$_4$NClO$_4$ (1.0 mmol), DMF (10 mL), graphite felt anode, copper plate cathode, CCE = 20 mA, 4 h (5.97 F/mol), 0 °C, $E/Z > 15/1$; DMF $N,N$-dimethylformamide, CCE constant current electrolysis. The percentage listed under each of the substrate refers to the isolated yield obtained in the reaction. $^a$Cs$_2$CO$_3$ (10 mol%) was used.

**Fig. 6 | Synthetic utility investigation. a** Gram-scale reaction. **b** Using solar cell as electricity supply. **c** Derivatization of the cross-coupling products. TSOH $p$-toluenesulfonic acid, NIS $N$-iodosuccinimide, DCM dichloromethane.

role of **cat 1** and **HT-1** in the reaction. Third, we conducted control experiments to elucidate the necessity of both electrode reactions (Fig. 7e). In a divided cell, no desired coupling product **3a** was observed (Eq (a)), while benzaldehyde was detected in the anodic chamber, and a possible allylbenzene carbanion (see Supplementary Fig. 32 for its ultraviolet−visible spectra) was proposed in the cathodic chamber. Further replacing **HT-1** with stoichiometric oxidant 2,2,6,6-tetramethyl-1-oxopiperidinium tetrafluoroborate

only afforded benzaldehyde (Eq (b)). These results exclude the mechanism which only relies on anodic oxidation. The kinetic isotope effect (KIE) study was also investigated with intramolecular competition and parallel experiment[57], and it reveals that the oxidation of **1a** is the rate-determining step (RDS, Eq (c)-(d)).

On the basis of the above experimental observations and related mechanism reports[40,47–49], a plausible reaction mechanism is proposed (Fig. 7f). Under anodic oxidation, pentacoordinated Co$^{II}$-salen (**cat 1**) is

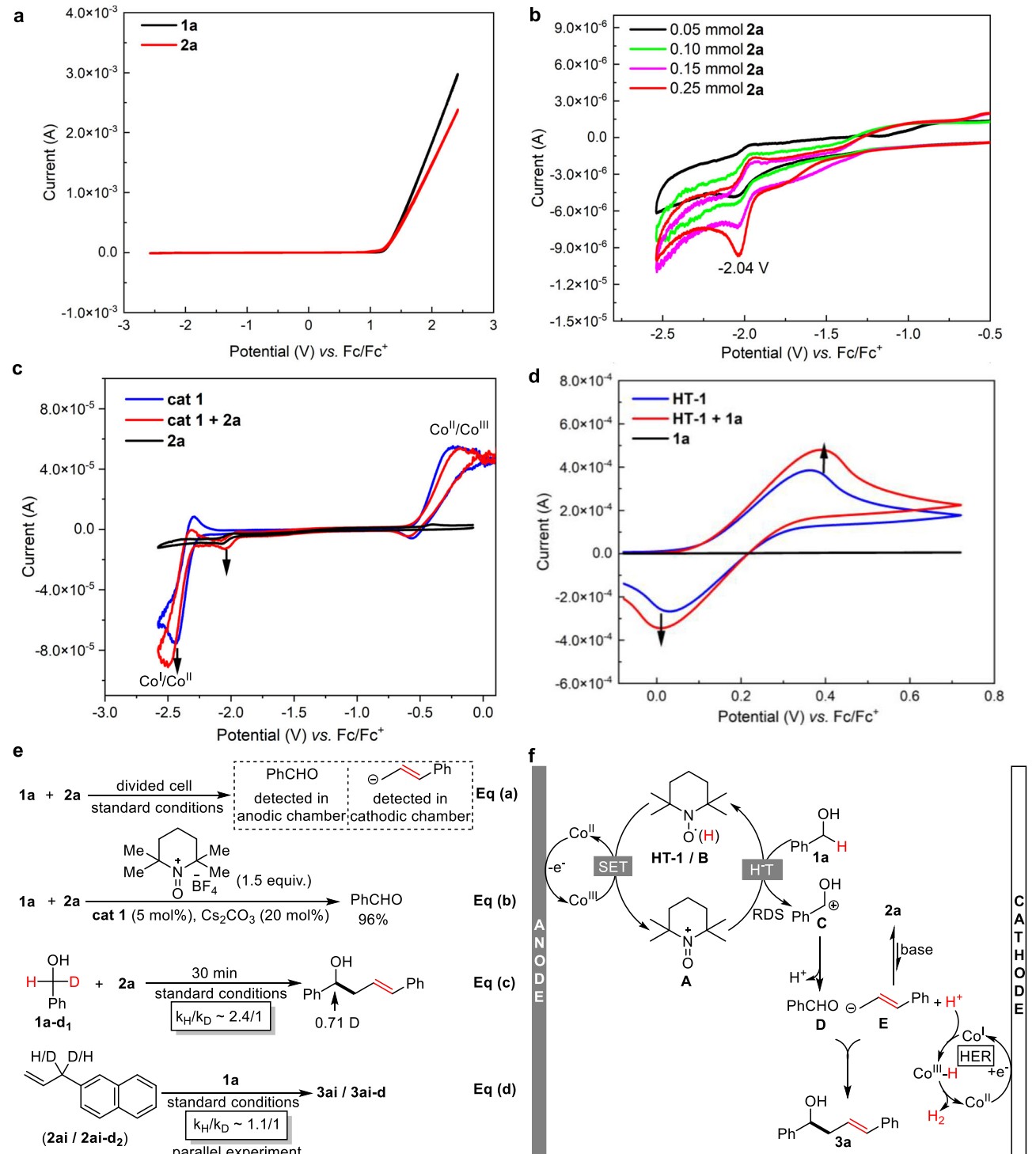

**Fig. 7 | Mechanism investigation. a** Cyclic voltammogram of **1a** (0.05 mmol) and **2a** (0.05 mmol) in DMF (2.5 mL) containing 0.05 mmol Cs$_2$CO$_3$. **b** Cyclic voltammogram of **2a** in DMF (2.5 mL) containing 0.05 mmol Cs$_2$CO$_3$. **c** Cyclic voltammogram of **cat 1** (0.01 mmol) and **2a** (0.3 mmol) in DMF (2.5 mL) containing 0.05 mmol Cs$_2$CO$_3$. **d** Cyclic voltammogram of TEMPO (**HT-1**) (0.05 mmol) and **1a** (0.1 mmol) in DMF (2.5 mL). **e** Control experiments. **f** Proposed reaction mechanism.

first oxidized to Co$^{III}$ species, which can subsequently initiate the oxidation of **HT-1** via single electron transfer (SET). The in situ generated reactive species **A** abstracts a hydride from **1a** to regenerate a protonated TEMPO (**B**) and deliver cation **C**, which rapidly converts to benzaldehyde **D** with the conjugation effect of the oxygen atom. On the other hand, the deprotonation of weakly acidic substrates is promoted by the Co$^{I}$ species arising at the cathode from **cat 1**, and position-isomerized carbanion **E** is simultaneously generated along

with hydrogen byproduct. Direct reaction between **D** and **E** affords the dehydrogenative product **3a**.

## Discussion

In summary, we have developed a paired electrocatalysis strategy leading to an unconventionally chemoselective cross-dehydrogenative coupling of alcohols with allylic and benzylic C-H bonds. The paired electrocatalysis consisting of H·T and HER catalysts enables a dual

activation for alcohols and weakly acidic C-H substrates, respectively. The success of this strategy largely relies on a pentacoordinated Co$^{II}$-salen HER catalyst, and it shows unmatched compatibility with the H$^-$T catalyst owing to its large redox-potential gap and suitable oxidative property. With the catalyst combinations, unconventional C(sp$^3$)-C(sp$^3$) coupling products are readily accessed along with hydrogen. The further application of the paired electrocatalysis strategy in challenging transformations has been actively explored in our laboratory.

## Methods
### General procedure for electrochemical CDC reaction
An undivided cell was equipped with a magnet stirrer, copper plate (1.8 *1.5 cm$^2$), and graphite felt (1.8 *1.5 cm$^2$), as cathode and anode, respectively (the electrolysis setup is shown in Supplementary Fig. 28). The substrate benzyl alcohol (52 μL, 0.5 mmol), allylbenzene **2a** (199 μL, 1.5 mmol), Cs$_2$CO$_3$ (31 mg, 0.1 mmol), **cat 1** (14 mg, 0.025 mmol), TEMPO (16 mg, 0.1 mmol) and $^n$Bu$_4$NClO$_4$ (341 mg, 1.0 mmol) were added to the solvent DMF (10 mL). The resulting mixture was allowed to stir and electrolyze under constant current conditions (20 mA, $J = 7.4$ mA•cm$^{-2}$) at 0 °C for 4 h. The reaction mixture was subsequently poured into water (100 mL) and extracted with ethyl acetate (40 mL × 3). The combined organic phases were washed with saturated brine solution (100 mL). The volatile solvent was then removed with a rotary evaporator, and the residue was purified by column chromatography (PE/EA = 8/1-5/1, v/v) on silica gel to afford the desired product **3a** (95 mg) in 85 % yield.

## Data availability
All data supporting the findings of this study, including experimental details, spectroscopic characterization data for all compounds are available in the text and the Supplementary Information section, or from the corresponding author upon request. Crystallographic data for the structure reported in this Article have been deposited at the Cambridge Crystallographic Data Centre, under deposition numbers CCDC 2313154 (**cat 1**). Copies of the data can be obtained free of charge via https://www.ccdc. cam.ac.uk/structures/.

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

## Acknowledgements

This work was supported by the National Natural Science Foundation of China (21702113 to S.Z., and 92061110 to M.-B.L.), the Anhui University (S020318006/069 to S.Z., and S020118002/113 to M.-B.L.), and the Anhui Provincial Natural Science Foundation (2308085Y14 to S.Z., and 2108085Y05 to M.-B.L.).

## Author contributions

S.Z. and M.-B.L. conceived the project, designed the experiments and wrote the manuscript. K.L., M.L., X.L. and X.Z. performed the experimental work. K.L. and M.L. contributed equally for this work. Refinement and analysis of single-crystal data was conducted by Y.Z. and W.F. All authors discussed the results and commented on the manuscript.

## Competing interests

The authors declare no competing interests.
