## [Peer Review File · Nature Communications]

Paired Electrocatalysis Unlocks Cross-Dehydrogenative Coupling of C(sp³)-H Bonds Using A Pentacoordinated Cobalt-Salen CatalystReviewers' Comments:

Reviewer #1:

Remarks to the Author:

The manuscript by Zhang and co-authors describes a paired electrolysis method for achieving double C(sp³)-H cross-coupling. The authors introduce HT (TEMPO) and HER (Co-salen) catalysts to enhance the reaction process. Broad group tolerance and high regio/site-selectivity are well presented by using synergistic effect of the catalyst combinations. Moreover, the bonding formation between C(sp³)-H bond adjacent to hydroxyl and C(sp³)-H of allyl is unexplored and useful for synthetic chemistry. Although the similar work which Co-salen catalyst is used to promote C(sp³)-H cross-coupling has been reported before (Xu et al, *Angew. Chem. Int. Ed.* 2022, 61, e202115954), the merits of this work are well performed with its mild condition and different compatibility.

Overall, I think this work is worth to be published in *Nat. Commun* after minor revision. To improve this work, there are some suggestions:

1. Table 3, 3cd: Can you find the homo-coupling product of cinnamyl alcohols?
2. Table 2, 3: The regioselectivity of some examples should be described scrupulously (3m, and so on).
3. SI spectra, the spectrum of 3j is missing.
4. Main text, line 174: "As shown in Figure 2A" is wrong.
5. Main text, line 178: "." need not be overstriking.
6. Figure 5c and 5d: The experiment of cat-1 with gradient 2a loading is necessary to prove catalytic current. TEMPO and 1a too.
7. Related reference about Cobalt-catalysed electrochemical C-H functionalization reactions should be cited (*J. Am. Chem. Soc.* 2017, 139, 18452–18455, *J. Am. Chem. Soc.* 2018, 140, 4195–4199)

Reviewer #2:

Remarks to the Author:

The authors do an excellent job describing their innovative, synthetically useful/valuable, and mechanistically interesting research. Their efforts led to the development of paired electrocatalysts to achieve cross dehydrogenative coupling of C(sp³)-H bonds. That the scope of the chemistry is broad is evident by the inclusion of more than 80 examples. The utility of the methodology is further witnessed by demonstrating that the chemistry can be carried out on gram-scale.

Once the authors have attended to the items listed below, I believe the manuscript will be suitable for publication. The science is interesting and innovative and merits publication.

- References to the publications of Aust need to be included when discussing paired electrolyses. She led pioneering efforts from BASF that led to landmark achievements. I also urge the inclusion of references Moeller's discussions dealing with paired electrolyses.
- The authors have written about some very important topics and have pointed out critical features of the proposed chemistry that must be fulfilled. For example: "Active anodic and cathodic catalysts have opposite redox nature, and their mutual interference would result in deteriorated catalytic-efficiency." I recommend that the sentence be rewritten. I am concerned for readers who might be unfamiliar with paired electrosynthesis. How about something like this?
Our plan requires two redox catalysts. One, call it M+•, is generated at the anode and is to be used to mediate oxidation. The other catalyst, C-•, is generated at the cathode and is to be used to mediate reduction. In principle, being of opposite polarity, the two might react with one another. For example, C-• might serve as a reducing agent that converts M+• to M. Alternatively, M+• might oxidize C-• to form C. Should these processes occur, the overall catalytic efficiency will be reduced.

- The authors have written: "also amenable to give the corresponding product 3f". Note that 3f is not a product. Rather, it is one of the SUBSTRATES shown in Table 2. Thus, Table 2 requires clarification in order to be clear that the percentages listed under each of the substrates refers to the yield obtained IN THE REACTION.
- In Table 2, the arrows that appear next to some of the structures e.g., the allylbenzenes, need to be defined.
- I agree that the results suggest that a radical pathway is not involved and that, as the authors have written "None of the ring expansion byproduct was detected, further confirming an ionic pathway." I have a slight problem with this, however, since the mass balance is < 70%. In principle, therefore, a rearrangement product might have been produced. While I believe the authors are correct, I urge them to exercise caution.
- I welcome use of the word "suggests" in the following text: "This result suggests that protocol not only provides a route for organic transformation but also enables an avenue for fuels." However, I once again urge that the authors be careful and avoid what I think is an unnecessary exaggeration when they attempt to tie the chemistry to fuels. Other researchers (e.g., Moeller and B. Nguyen) have actually harnessed the hydrogen produced at the cathode to carry out hydrogenation reactions in a separate cell after using a cannula to achieve the transfer.
- I have similar negative thoughts when the authors conclude: "with the clean fuel hydrogen. We believe that this paired electrocatalysis strategy would open up an appealing platform that merges synthetic chemistry with energy chemistry."
- I looked at the supplementary materials, being particularly interested in a few items. Thus, when I read "Using a simple solar cell as electricity supply largely" in the text, I wondered about the specifics ... what was meant by "simple solar cell". The picture shown in Figure S24 is not helpful as it lacks detail. For example, is it a single solar panel? What size is it? What company was it purchased from, etc.?
- One reads: "while benzaldehyde and allylbenzene carbanion (see Fig. S26 for its UV-vis spectra)". The control cited in the supplementary section viz., the use of BuLi to generate the carbanion, seems reasonable. Has the spectrum of the authentic carbanion been reported in the literature?
- Change "bonds" to "bond" in: "for C(sp³)-C(sp³) bonds construction"
- Change "the" to "a" in: "rely on the single activation mode"
- Change "We herein developed a paired electrocatalysis strategy to access" to "Herein we describe our development of a paired electrocatalysis strategy to access"
- Change "catalyst pentacoordinated Co-salen was disclosed, and it displayed a large redox-potential" to read: "catalyst pentacoordinated Co-salen is disclosed. The catalyst displayed a large redox-potential"
- Change "is" to "was" in: "A strategy based on cross-dehydrogenative coupling (CDC) is first unveiled by Li"
- Change "would" to "could" in: "which would result in some selectivity"
- Change "approaches only involve the activation of anodic oxidation" to read: "approaches only involve anodic oxidation"

- Insert "of" before "alcohols" in: "The cross-dehydrogenative coupling alcohols with"
- Change "direct approach upgrading simple alcohols to value-added ones" to read: "direct approach to upgrade simple alcohols to value-added alcohols"
- More reactive toward what? Please specify. I refer to: "O-H is more reactive than C-H bonds"
- Change "property" to "properties" in: "relationship of structure and redox property"
- Change "identified" to "established" in: "was unambiguously identified by the single crystal diffraction" and delete "the"
- Change "We concluded all of" to read: ""We recorded and identified all of"
- What does "improve" mean in the following context? "improve the oxidation potential of CoII/CoIII with the most..." Please be clearer.
- Specify where the following is illustrated in the text. I refer to: "was demonstrated by accelerating the catalytic cycle of TEMPO at lower anodic"
- The expression "isomerizing-selectivity" is unclear. I refer to: "with exclusive C-C coupling and isomerizing-selectivity"
- Larger than what? And lower than what? Be careful as these words are ambiguous. I refer to: "catalyst with larger potential gap and H-T catalyst with lower oxidation peak would benefit the"
- Be specific. That is, specify the diaryl ketone to which you are referring in: "Moreover, diaryl ketone was"
- The wording "which conventionally preferred to proceed a Pinacol coupling" is cumbersome and confusing. A pinacol coupling of $\text{PhCOC}_6\text{H}_4\text{CH}_3$?
- The following needs to be reworded "homoallylic alcohol 3l proceeded a facile dehydration to give a conjugated"
- Recommend changing "series of cyclic voltametric experiment and" to read: "series of cyclic voltammograms and"
- The authors write: "1a and 2a seem to be redox inert within". Seem? They either are or are not. Which is it?
- Change "excessive" to "excess" in: "Treatment of cat 1 with excessive 2a led to significant"
- The authors write: "In a divided cell, no desired coupling product". It is not uncommon for materials to migrate from one to the other cell compartment. Did this occur?
- Change "was" to "is" in: "plausible reaction mechanism was proposed"
- Change "could" to "can" in: "which could subsequently initiate"
- Change to "regenerate" in: "deliver cation C and regenerates a protonated"
- Be careful with "Over the cathode surface" fore it is possible that the chemistry could occur in the double layer or in the bulk rather than at the surface.

- Is a proton dissociated? I don't think this is the proper/accurate word. I refer to: "the dissociation of proton from weakly acidic"
- Change "arising from cat 1, and isomerized" to read: "arising at the cathode from cat 1, and isomerized"
- Insert "have" following "we" in: "In summary, we develop a paired electrocatalysis"
- Change "electrocatalysis strategy for an unconventionally" to read: "electrocatalysis strategy leading to an unconventionally"
- Change to "alcohols" in: "alcohol and weakly acidic"
- It would be preferable to write: "available in the text and the Supporting Information section," rather than: "available within the paper and its Supplementary Methods"
- Delete "reasonable" from "the corresponding author upon reasonable request"

Reviewer #3:

Remarks to the Author:

Zhang and Li describe an interesting paired electrocatalysis strategy to access C(sp³)-C(sp³) coupling reactions between two distinct C(sp³)-H bonds. The catalyst combination consists of a hydride transfer mediator TEMPO and a "HER" catalyst, and they could selectively promote alcohol oxidation and the reduction of weakly acidic C-H bonds, respectively. Moreover, they devise a pentacoordinated Co-salen catalyst with suitable redox properties to achieve the synergistic effect of paired electrocatalysis. This protocol provides an efficient solution for conventionally challenging transformations. Hence, publication in Nature Communications is endorsed pending minor revision.

1. In the depicted mechanism (Figure 5f), the carbocation C adjacent to the hydroxy group should preferentially isomerize into an aldehyde. Additionally, benzaldehyde was also detected in the control experiment. The benzaldehyde should be depicted as an intermediate in the reaction.
2. For the divided cell experiment, the author mentioned "while benzaldehyde and allylbenzene carbanion (see Fig. S26 for its UV-vis spectra) were detected in anodic and cathodic chamber, respectively." In fact, the UV-Vis spectrum cannot detect benzaldehyde or allylbenzene carbanion, only signals can be detected to speculate on possible species. Therefore, the description here is inappropriate. Please also modify the corresponding expressions in the SI.
3. The steric variations on the cobalt-salen catalyst are recommended to be explored to better understand the structure-activity relationship of the catalysts.
4. Some extra nitroxyl-radical mediators such as N-hydroxyphthalimide, 4-oxo-tempo, and 4-methoxy-tempo should be tested in the reaction further to illustrate the synergistic effect of the catalyst combinations.
5. I wonder whether this protocol is applicable to other substrates bearing weakly acidic C(sp³)-H bonds, and carbocation precursors other than alcohols. Please provide some examples of electron-donor toluene derivatives for Table 2, or give some comments.
6. The use of a perchlorate electrolyte is in fact not desirable from a safety and environmental point of view. Perchlorate electrolytes are in fact highly energetic and are toxic as they are thyroid hormone disruptors (see DOI: 10.1021/acs.oprd.2c00111). The authors in fact had safer alternative electrolytes with either Bu₄NBF₄ or Bu₄NPF₆ (Table S1, entries 2 and 3) but unfortunately did not choose these for their reaction scope. The authors need to add a safety warning regarding the use of

tetrabutylammonium perchlorate electrolyte.

7. For all Schemes and tables involving electrochemical synthesis, could the author inform the number of F/mol? This is helpful for the readers, as it allows to quickly access the Faradaic efficiency.

8. Compound 3g needs to be purified.

9. There are some typos in the manuscript. The author should check the manuscript and supporting information to ensure that there are no similar errors.

In page 6, line 97, it is suggested to change "Entry" to "Entries".

Line 65, Page 5, "chemoselectivty" should be "chemoselectivity".

Line 149, Page 10, "byproduct" should be "byproducts".

Line 173, Page 12, "voltametric" should be "voltammetric".

Response to the reviewer's comments

Manuscript number: NCOMMS-23-63902

Title: " Paired Electrocatalysis Unlocks Cross-Dehydrogenative Coupling of C(sp³)-H Bonds Using A Pentacoordinated Cobalt-Salen Catalyst"

We thank the three reviewers who took part in the revision of the manuscript for carefully revising the manuscript and for their thoughtful comments and suggestions which have helped to improve the quality of the manuscript.

Reviewer 1

The manuscript by Zhang and co-authors describes a paired electrolysis method for achieving double C(sp³)-H cross-coupling. The authors introduce HT (TEMPO) and HER (Co-salen) catalysts to enhance the reaction process. Broad group tolerance and high regio/site-selectivity are well presented by using synergistic effect of the catalyst combinations. Moreover, the bonding formation between C(sp³)-H bond adjacent to hydroxyl and C(sp³)-H of allyl is unexplored and useful for synthetic chemistry. Although the similar work which Co-salen catalyst is used to promote C(sp³)-H cross-coupling has been reported before (Xu et al, Angew. Chem. Int. Ed. 2022, 61, e202115954), the merits of this work are well performed with its mild condition and different compatibility.

Overall, I think this work is worth to be published in Nat. Commun after minor revision. To improve this work, there are some suggestions:

Response: We thank the reviewer for the kind words and thoughtful comments on our work.

1. Table 3, 3cd: Can you find the homo-coupling product of cinnamyl alcohols?

Response: We thank the reviewer for this comment. We carefully conducted the reaction again, and only unreacted alcohol and the aldehyde intermediate arising from it were detected in the reaction mixture. The steric hindrance of the substrate might attribute to the less reactivity.

2. Table 2, 3: The regioselectivity of some examples should be described scrupulously (3m, and so on).

Response: We thank the reviewer for this insightful comment, and we fully agree with this point. Corresponding comments have been added to the manuscript.

3. SI spectra, the spectrum of 3j is missing.

Response: We thank the reviewer for this comment, which has helped to improve the quality of the manuscript. Although the spectra of product 3j and 3k are same to that of 3a and 3ad, their ¹H NMR spectra have been included in in the SI (page S57-S58).

4. Main text, line 174: "As shown in Figure 2A" is wrong.

Response: We thank the reviewer for this comment, and we are sorry for this mistake. "Figure 2A" has been corrected to "Fig. 5A".

5. Main text, line 178: "." need not be overstriking.

Response: We thank the reviewer for this comment, and we are sorry for this mistake. We have

corrected it.

6. Figure 5c and 5d: The experiment of cat-1 with gradient 2a loading is necessary to prove catalytic current. TEMPO and 1a too.

Response: We thank the reviewer for this insightful comment, and we fully agree with this point. The titration experiments have been conducted and included in the SI (page S14-S15). As shown below, the catalytic current increases upon increasing the loading of substrates **2a** and **1a**, further verifying the catalytic role of both **cat 1** and TEMPO.

7. Related reference about Cobalt-catalysed electrochemical C-H functionalization reactions should be cited (J. Am. Chem. Soc. 2017, 139, 18452–18455, J. Am. Chem. Soc. 2018, 140, 4195–4199)

Response: We thank the reviewer for this comment. The references have been added to the manuscript (ref 54-55).

Reviewer 2

The authors do an excellent job describing their innovative, synthetically useful/valuable, and mechanistically interesting research. Their efforts led to the development of paired electrocatalysts to achieve cross dehydrogenative coupling of C(sp³)-H bonds. That the scope of the chemistry is broad is evident by the inclusion of more than 80 examples. The utility of the methodology is further witnessed by demonstrating that the chemistry can be carried out on gram-scale.

Once the authors have attended to the items listed below, I believe the manuscript will be suitable for publication. The science is interesting and innovative and merits publication.

Response: We thank the reviewer for the kind words and thoughtful comments on our work.

- References to the publications of Aust need to be included when discussing paired electrolyses. She led pioneering efforts from BASF that led to landmark achievements. I also urge the inclusion of references Moeller's discussions dealing with paired electrolyses.

Response: We thank the reviewer for the comment, and we fully agree with this point. The references have been included in the manuscript (ref 30-32).

- The authors have written about some very important topics and have pointed out critical features of the proposed chemistry that must be fulfilled. For example: “Active anodic and cathodic catalysts have opposite redox nature, and their mutual interference would result in deteriorated catalytic-efficiency.” I recommend that the sentence be rewritten. I am concerned for readers who might be unfamiliar with paired electrosynthesis. How about something like this?

Our plan requires two redox catalysts. One, call it $M^{+\bullet}$, is generated at the anode and is to be used to mediate oxidation. The other catalyst, $C^{-\bullet}$, is generated at the cathode and is to be used to mediate reduction. In principle, being of opposite polarity, the two might react with one another. For example, $C^{-\bullet}$ might serve as a reducing agent that converts $M^{+\bullet}$ to M . Alternatively, $M^{+\bullet}$ might oxidize $C^{-\bullet}$ to form C . Should these processes occur, the overall catalytic efficiency will be reduced.

Response: We really appreciate the reviewer for this insightful suggestion, and we fully agree with this point. The description has been adopted (Fig.2c) and added to the manuscript.

The authors have written: “also amenable to give the corresponding product 3f”. Note that 3f is not a product. Rather, it is one of the SUBSTRATES shown in Table 2. Thus, Table 2 requires clarification in order to be clear that the percentages listed under each of the substrates refers to the yield obtained IN THE REACTION.

Response: We thank the reviewer for the comment, and we fully agree with this point. The clarification has been added to the table footnote.

- In Table 2, the arrows that appear next to some of the structures e.g., the allylbenzenes, need to be defined.

Response: We thank the reviewer for the comment, and we fully agree with this point. The clarification has been added to the table.

- I agree that the results suggest that a radical pathway is not involved and that, as the authors have written “None of the ring expansion byproduct was detected, further confirming an ionic pathway.” I have a slight problem with this, however, since the mass balance is $< 70\%$. In principle, therefore, a rearrangement product might have been produced. While I believe the authors are correct, I urge them to exercise caution.

Response: We thank the reviewer for the comment. However, we detected the reaction mixture with GC-MS and TLC, and only some unreacted substrates and aldehyde intermediates were observed. Nevertheless, the expression has been modified to “The possible radical-initiated ring expansion byproducts were not detected in the reaction mixture, further suggesting that the radical pathway might not involve in the transformation”.

- I welcome use of the word “suggests” in the following text: “This result suggests that protocol not only provides a route for organic transformation but also enables an avenue for fuels.” However, I once again urge that the authors be careful and avoid what I think is an unnecessary exaggeration when they attempt to tie the chemistry to fuels. Other researchers (e.g., Moeller and B. Nguyen) have actually harnessed the hydrogen produced at the cathode to carry out hydrogenation reactions in a separate cell after using a cannula to achieve the transfer.

- I have similar negative thoughts when the authors conclude: “with the clean fuel hydrogen. We believe that this paired electrocatalysis strategy would open up an appealing platform that merges synthetic chemistry with energy chemistry.”

Response: We thank the reviewer for the above comments, and we fully agree with this point. The description has been modified, and the “fuel” has been deleted.

- I looked at the supplementary materials, being particularly interested in a few items. Thus, when I read “Using a simple solar cell as electricity supply largely” in the text, I wondered about the specifics ... what was meant by “simple solar cell”. The picture shown in Figure S24 is not helpful as it lacks detail. For example, is it a single solar panel? What size is it? What company was it purchased from, etc.?

Response: We thank the reviewer for the comment, and we fully agree with this point. The details of solar cell have been added to the SI.

- One reads: “while benzaldehyde and allylbenzene carbanion (see Fig. S26 for its UV-vis spectra)”. The control cited in the supplementary section viz., the use of BuLi to generate the carbanion, seems reasonable. Has the spectrum of the authentic carbanion been reported in the literature?

Response: We thank the reviewer for the comment. The reference for the carbanion spectrum has been included in the SI.

- Change “bonds” to “bond” in: “for C(sp³)-C(sp³) bonds construction”

Response: We thank the reviewer for the comment. We have corrected it.

- Change “the” to “a” in: “rely on the single activation mode”

Response: We thank the reviewer for the comment. We have corrected it.

- Change “We herein developed a paired electrocatalysis strategy to access” to “Herein we describe our development of a paired electrocatalysis strategy to access”

Response: We thank the reviewer for the comment. We have corrected it.

- Change “catalyst pentacoordinated Co-salen was disclosed, and it displayed a large redox-potential” to read: “catalyst pentacoordinated Co-salen is disclosed. The catalyst displayed a large redox-potential”

Response: We thank the reviewer for the comment. We have corrected it.

- Change “is” to “was” in: “A strategy based on cross-dehydrogenative coupling (CDC) is first unveiled by Li”

Response: We thank the reviewer for the comment. We have corrected it.

- Change “would” to “could” in: “which would result in some selectivity”

Response: We thank the reviewer for the comment. We have corrected it.

- Change “approaches only involve the activation of anodic oxidation” to read: “approaches only involve anodic oxidation”

Response: We thank the reviewer for the comment. We have corrected it.

- Insert “of” before “alcohols” in: “The cross-dehydrogenative coupling alcohols with”

Response: We thank the reviewer for the comment. We have corrected it.

- Change “direct approach upgrading simple alcohols to value-added ones” to read: “direct approach to upgrade simple alcohols to value-added alcohols”

Response: We thank the reviewer for the comment. We have corrected it.

- More reactive toward what? Please specify. I refer to: “O-H is more reactive than C-H bonds”

Response: We thank the reviewer for the comment, and we fully agree with this point. We have modified the description to “O-H is more acidic than C-H bond and easily converted to an oxygen anion nucleophile”.

- Change “property” to “properties” in: “relationship of structure and redox property”

Response: We thank the reviewer for the comment. We have corrected it.

- Change “identified” to “established” in: “was unambiguously identified by the single crystal diffraction” and delete “the”

Response: We thank the reviewer for the comment. We have corrected it.

- Change “We concluded all of” to read: ““We recorded and identified all of”

Response: We thank the reviewer for the comment. We have corrected it.

- What does “improve” mean in the following context? “improve the oxidation potential of CoII/CoIII with the most...” Please be clearer.

Response: We thank the reviewer for the comment. The description has been modified to “CF₃ group was found to significantly improve the antioxidation stability of **cat 1** with the most positive peak at -0.157 V (vs. Fc/Fc⁺)”.

- Specify where the following is illustrated in the text. I refer to: “was demonstrated by accelerating the catalytic cycle of TEMPO at lower anodic”

Response: We thank the reviewer for the comment. The description has been modified to “The synergistic effect of the catalyst combination was demonstrated by the acceleration effect of **cat 1** on the oxidation of TEMPO”.

- The expression “isomerizing-selectivity” is unclear. I refer to: “with exclusive C-C coupling and isomerizing-selectivity”

Response: We thank the reviewer for the comment. We have specified the expression to “C=C position isomerizing selectivity”

- Larger than what? And lower than what? Be careful as these words are ambiguous. I refer to: “catalyst with larger potential gap and H-T catalyst with lower oxidation peak would benefit the”

Response: We thank the reviewer for the comment, and we fully agree with this point. To avoid

the ambiguous expression, we have corrected the description.

- Be specific. That is, specify the diaryl ketone to which you are referring in: “Moreover, diaryl ketone was”

Response: We thank the reviewer for the comment. We have specified the expression to “Moreover, 4-methylbenzophenone was”

- The wording “which conventionally preferred to proceed a Pinacol coupling” is cumbersome and confusing. A pinacol coupling of PhCOC₆H₄CH₃?

Response: We thank the reviewer for the comment. We have specified the expression to “which conventionally preferred to proceed a Pinacol coupling of 4-methylbenzophenone”

- The following needs to be reworded “homoallylic alcohol **3I** proceeded a facile dehydration to give a conjugated”

Response: We thank the reviewer for the comment. We have changed the expression to “the dehydration of homoallylic alcohol **3I** gives a conjugated light-emitting molecule **4**”.

- Recommend changing “series of cyclic voltametric experiment and” to read: “series of cyclic voltammograms and”

Response: We thank the reviewer for the comment. We have corrected it.

- The authors write: “**1a** and **2a** seem to be redox inert within”. Seem? They either are or are not. Which is it?

Response: We thank the reviewer for the comment. We have specified the expression to “both **1a** and **2a** seem to be redox inert within the window of -2.5-2.5 V (vs. Fc/Fc⁺)”

- Change “excessive” to “excess” in: “Treatment of cat **1** with excessive **2a** led to significant”

Response: We thank the reviewer for the comment. We have corrected it.

- The authors write: “In a divided cell, no desired coupling product”. It is not uncommon for materials to migrate from one to the other cell compartment. Did this occur?

Response: We thank the reviewer for the comment. As mentioned in the SI, we premixed the substrates **1a** and **2a**, and introduced the mixture to the anodic and cathodic chambers.

- Change “was” to “is” in: “plausible reaction mechanism was proposed”

Response: We thank the reviewer for the comment. We have corrected it.

- Change “could” to “can” in: “which could subsequently initiate”

Response: We thank the reviewer for the comment. We have corrected it.

- Change to “regenerate” in: “deliver cation C and regenerates a protonated”

Response: We thank the reviewer for the comment. We have corrected it.

- Be careful with “Over the cathode surface” fore it is possible that the chemistry could occur in

the double layer or in the bulk rather than at the surface.

Response: We thank the reviewer for the comment. We have corrected it.

- Is a proton dissociated? I don't think this is the proper/accurate word. I refer to: "the dissociation of proton from weakly acidic"

Response: We thank the reviewer for the comment. We changed the expression to "the deprotonation of weakly acidic substrates".

- Change "arising from cat 1, and isomerized" to read: "arising at the cathode from cat 1, and isomerized"

Response: We thank the reviewer for the comment. We have corrected it.

- Insert "have" following "we" in: "In summary, we develop a paired electrocatalysis"

Response: We thank the reviewer for the comment. We have corrected it.

- Change "electrocatalysis strategy for an unconventionally" to read: "electrocatalysis strategy leading to an unconventionally"

Response: We thank the reviewer for the comment. We have corrected it.

- Change to "alcohols" in: "alcohol and weakly acidic"

Response: We thank the reviewer for the comment. We have corrected it.

- It would be preferable to write: "available in the text and the Supporting Information section," rather than: "available within the paper and its Supplementary Methods"

Response: We thank the reviewer for the comment. We have corrected it.

- Delete "reasonable" from "the corresponding author upon reasonable request"

Response: We thank the reviewer for the comment. We have corrected it.

Reviewer 3

Zhang and Li describe an interesting paired electrocatalysis strategy to access C(sp³)-C(sp³) coupling reactions between two distinct C(sp³)-H bonds. The catalyst combination consists of a hydride transfer mediator TEMPO and a "HER" catalyst, and they could selectively promote alcohol oxidation and the reduction of weakly acidic C-H bonds, respectively. Moreover, they devise a pentacoordinated Co-salen catalyst with suitable redox properties to achieve the synergistic effect of paired electrocatalysis. This protocol provides an efficient solution for conventionally challenging transformations. Hence, publication in Nature Communications is endorsed pending minor revision.

Response: We thank the reviewer for the kind words and thoughtful comments on our work.

1. In the depicted mechanism (Figure 5f), the carbocation C adjacent to the hydroxy group should preferentially isomerize into an aldehyde. Additionally, benzaldehyde was also detected in the control experiment. The benzaldehyde should be depicted as an intermediate in the reaction.

Response: We thank the reviewer for this insightful comment, and we fully agree with this point. We have corrected the reaction mechanism.

2. For the divided cell experiment, the author mentioned “while benzaldehyde and allylbenzene carbanion (see Fig. S26 for its UV-vis spectra) were detected in anodic and cathodic chamber, respectively.” In fact, the UV-Vis spectrum cannot detect benzaldehyde or allylbenzene carbanion, only signals can be detected to speculate on possible species. Therefore, the description here is inappropriate. Please also modify the corresponding expressions in the SI.

Response: We thank the reviewer for the comment, and we fully agree with this point. The expression has been modified.

3. The steric variations on the cobalt-salen catalyst are recommended to be explored to better understand the structure-activity relationship of the catalysts.

Response: We thank the reviewer for the comment. The 3,5-di-tert-butyl-2-hydroxybenzaldehyde derived cobalt-salen catalyst (**cat 8**) has been investigated and included in the manuscript (Fig 3a, Table 1) and the SI (page S7). It showed that the steric hindrance slightly improves the potential gap of the catalyst (1.40 V).

4. Some extra nitroxyl-radical mediators such as N-hydroxyphthalimide, 4-oxo-tempo, and 4-methoxy-tempo should be tested in the reaction further to illustrate the synergistic effect of the catalyst combinations.

Response: We thank the reviewer for the comment. These nitroxyl-radical mediators have been explored and included in the SI (Table S2). Their CV have also been included in the SI (page S10-s11). These results further verify that the high oxidative potential would lead to deteriorated yield.

5. I wonder whether this protocol is applicable to other substrates bearing weakly acidic C(sp³)-H bonds, and carbocation precursors other than alcohols. Please provide some examples of electron-donor toluene derivatives for Table 2, or give some comments.

Response: We thank the reviewer for this insightful comment. Substrate (methylsulfinyl)benzene bearing weakly acidic C-H bond was tested with our protocol, and corresponding product **3I** was obtained in 21% yield. We also tried to replace alcohol with benzyl amine, whereas we only detected the product **3a** arising from the overoxidation, suggesting that more selective hydride transfer mediator is required. The electron rich toluene (4-methylanisole) was also tested in our protocol, and it failed to give the desired product (listed on the Table 2). Corresponding comments have been added to the manuscript. We are now trying developing other catalyst combinations to tolerate with these failed substrates.

6. The use of a perchlorate electrolyte is in fact not desirable from a safety and environmental point of view. Perchlorate electrolytes are in fact highly energetic and are toxic as they are thyroid hormone disruptors (see DOI: 10.1021/acs.oprd.2c00111). The authors in fact had safer alternative electrolytes with either Bu₄NBF₄ or Bu₄NPF₆ (Table S1, entries 2 and 3) but unfortunately did not chose these for their reaction scope. The authors need to add a safety warning regarding the use of tetrabutylammonium perchlorate electrolyte.

Response: We thank the reviewer for the comment, and we fully agree with this point. The safety warning has been added to the SI (page S18).

7. For all Schemes and tables involving electrochemical synthesis, could the author inform the number of F/mol? This is helpful for the readers, as it allows to quickly access the Faradaic efficiency.

Response: We thank the reviewer for the comment. The information has been provided in the footnote of Table 1-3.

8. Compound 3g needs to be purified.

Response: We thank the reviewer for the comment, which have helped to improve the quality of the manuscript. The spectra of **3g** have been updated.

9. There are some typos in the manuscript. The author should check the manuscript and supporting information to ensure that there are no similar errors.

In page 6, line 97, it is suggested to change "Entry" to "Entries".

Line 65, Page 5, "chemoselectivty" should be "chemoselectivity".

Line 149, Page 10, "byproduct" should be "byproducts".

Line 173, Page 12, "voltametric" should be "voltammetric".

Response: We thank the reviewer for the comment, which have helped to improve the quality of the manuscript. All of the typos have been corrected, and we also double-checked our manuscript and SI.

Reviewers' Comments:

Reviewer #1:

Remarks to the Author:

After reading through the author's response to the critiques, I do believe that the manuscript does now reach the level needed for publication in Nature Communications. Therefore, I recommend publication.

Reviewer #2:

Remarks to the Author:

The authors have done an excellent job responding to my previously voiced concerns/suggestions. A few minor items remain, viz.,

- It looks like the gap is smaller rather than larger. I am referring to: "pentacoordinated catalysts (cat 1-cat 3, 1.98-1.92 V) have uniformly larger redox potential gap than that of the conventional catalysts cat 4-cat 8 (1.14-1.40 V)."
- No need to capitalize "pinacol" in: "proceed a Pinacol coupling of 4-methylbenzophenone"
- Remove "was" from "(methylsulfinyl)benzene was proved to be suitable to give the desired"
- This is still a bit challenging to visualize – without making a mistake. To clarify, please show one example of a transformation where the double bond has moved and the coupling site defined explicitly. I am referring to "Specifically, both allyl groups underwent isomerization to give" and the arrows that appear in Table 2.

Reviewer #3:

Remarks to the Author:

Previous comments have been properly addressed. This work is recommended for publication with Nature Communications.

Response to the reviewer's comments

We thank the three reviewers who took part in the revision of the manuscript for carefully revising the manuscript and for their thoughtful comments and suggestions which have helped to improve the quality of the manuscript.

Reviewer #1 (Remarks to the Author):

After reading through the author's response to the critiques, I do believe that the manuscript does now reach the level needed for publication in Nature Communications. Therefore, I recommend publication.

Response: We thank the reviewer for the kind words.

Reviewer #2 (Remarks to the Author):

The authors have done an excellent job responding to my previously voiced concerns/suggestions. A few minor items remain, viz.,

Response: We thank the reviewer for the kind words.

- It looks like the gap is smaller rather than larger. I am referring to: "pentacoordinated catalysts (cat 1-cat 3, 1.98-1.92 V) have uniformly larger redox potential gap than that of the conventional catalysts cat 4-cat 8 (1.14-1.40 V)."

Response: We thank the reviewer for this comment, but the reviewer may have misunderstood the meaning. The potential gap data (1.98-1.92 V, 1.14-1.40 V) is calculated by subtraction between the anodic potential peak ($E_{1/2}\text{Co}^{\text{II/III}}$) and the cathodic peak ($E_{1/2}\text{Co}^{\text{III/II}}$). Consequently, **cat 1-cat 3** should have larger redox potential gap than catalysts **cat 4-cat 8**.

- No need to capitalize "pinacol" in: "proceed a Pinacol coupling of 4-methylbenzophenone"

Response: We thank the reviewer for this comment. It has been fixed.

- Remove "was" from "(methylsulfinyl)benzene was proved to be suitable to give the desired"

Response: We thank the reviewer for this comment. It has been fixed.

- This is still a bit challenging to visualize – without making a mistake. To clarify, please show one example of a transformation where the double bond has moved and the coupling site defined explicitly. I am referring to "Specifically, both allyl groups underwent

isomerization to give" and the arrows that appear in Table 2.

Response: We thank the reviewer for this comment which have helped to improve the quality of the manuscript. The structure of product **3s** has been included in the Table 2 as an example.

Reviewer #3 (Remarks to the Author):

Previous comments have been properly addressed. This work is recommended for publication with Nature Communications.

Response: We thank the reviewer for the kind words.